# Comparison of Adalimumab to Other Targeted Therapies in Rheumatoid Arthritis: Results from Systematic Literature Review and Meta-Analysis

**DOI:** 10.3390/jpm12030353

**Published:** 2022-02-25

**Authors:** Fabio Cacciapaglia, Vincenzo Venerito, Stefano Stano, Marco Fornaro, Giuseppe Lopalco, Florenzo Iannone

**Affiliations:** Rheumatology Unit, Department of Emergency and Organs Transplantation (DETO), University of Bari, 70124 Bari, Puglia, Italy; vincenzo.venerito@uniba.it (V.V.); stano.stefano.5@gmail.com (S.S.); marco.fornaro@uniba.it (M.F.); giuseppe.lopalco@uniba.it (G.L.); florenzo.iannone@uniba.it (F.I.)

**Keywords:** efficacy, safety, Bayesian meta-analysis, rheumatoid arthritis, targeted therapies

## Abstract

Few studies compared adalimumab to other targeted therapies in head-to-head randomized clinical trials (RCTs) for rheumatoid arthritis (RA), but multiple comparisons are not available. This Bayesian Network Meta-Analysis evaluated which targeted therapy is more likely to achieve ACR50 response with good safety at 24 weeks of treatment in RA. A systematic literature review was conducted for head-to-head phase 3 RCTs that compared adalimumab to other targeted therapies in combination with methotrexate (MTX) or as monotherapy to treat RA patients, and searched through MEDLINE, EMBASE, Cochrane Library and Clinicaltrial.gov. The outcomes of interest were ACR50 response and withdrawals due to adverse events at 24 weeks. WinBUGS 1.4 software (MRC Biostatistics Unit, Cambridge, UK) was used to perform the analyses, using a random effect model. Sixteen studies were included in the analysis. The most favorable SUCRA for the ACR50 response rate at 24 weeks of treatment in combination with MTX was ranked by upadacitinib, followed by baricitinib, tofacitinib and filgotinib. As monotherapy, the highest probability was ranked by tocilizumab followed by sarilumab. No significant differences in safety profile among treatment options were found. Jak-inhibitors in combination with MTX and interleukin-6 antagonism as monotherapy showed the highest probability to achieve ACR50 response after 24 weeks of treatment. None of assessed targeted therapies were associated to risk of withdrawal due to adverse events. Key messages: Direct and indirect comparison between adalimumab and other targeted therapies demonstrated some differences in terms of efficacy that may help to drive RA treatment. Jak-inhibitors and interleukine-6 antagonists ranked as first in the probability to achieve ACR50 response after 24 weeks of treatment in combination with methotrexate or monotherapy, respectively.

## 1. Introduction

Rheumatoid arthritis (RA) is a chronic inflammatory disease that leads to joint damage and disability if not promptly treated with effective agents [1]. Over the last two decades, RA treatment has been implemented by different therapeutic strategies, with multi-target options such as specific interleukins inhibition or cellular targets [2,3]. More recently, the discovery of agents targeting intracellular pathways downstream of cell receptor activation also broadened the array of available treatment alternatives [4]. All these innovative drugs have shown efficacy and safety when administered to RA patients with active disease and inadequate response to conventional synthetic disease modifying anti-rheumatic drugs (csDMARDs), and EULAR recommend the access to multiple drugs with different mechanisms of action to address the heterogeneity of RA according to disease activity, safety issues, structural damage and comorbidities [5].

Adalimumab is a monoclonal anti-tumor necrosis factor human antibody, considered the blockbuster drug for RA treatment, and the only one that has been compared to other different biologic (b) or target synthetic (ts) DMARDs, both in monotherapy and combination with methotrexate (MTX), in head-to-head randomized clinical trials (RCTs) with some evidence of superiority [6]. However, no data on multiple comparisons are available [7]. In absence of head-to-head RCTs, required to estimate which treatment is the most effective in comparison to another drug, indirect comparisons that use a common comparator may also be useful. All available data from both direct and indirect comparisons are the essence of mixed treatment comparison (MTC), as an extension of meta-analysis, which allows multiple pairwise comparisons to be made across a range of different treatments in a Bayesian framework [8,9]. On the other hand, the availability of biosimilar agents had led clinicians to choose drugs based mainly on the principle of non-inferiority effect and sustainability rather than the most effective agent [10], so the best RA treatment remains a clinically unmet need [11].

Therefore, we performed a Bayesian Network meta-analysis of all RCTs that explored the relative efficacy and safety of different targeted therapies compared to adalimumab in achieving the 50% American College of Rheumatology response (ACR50) after 24 weeks of treatment in RA patients, with previous ineffective experience to MTX.

## 2. Methods

### 2.1. Data Source, Studies Selection and Data Extraction

An extensive literature search was carried out in accordance with the Preferred Reporting Items for Systematic Reviews and Meta-analyses (PRISMA) [12] statement for transparent reporting of systematic reviews and meta-analysis to identify results of RCTs that evaluated clinically biologic agents and small molecules at licensed doses to treat RA. We performed systematic reviews of MEDLINE, EMBASE and Cochrane Library databases, and Clinicaltrials.gov searching for all published phase 3 RCTs up to June 2021.

Any published study comparing adalimumab to other targeted therapies in patients with active RA and inadequate responders to previous csDMARDs for at least 3 months was included. More specifically, only studies comparing adalimumab or adalimumab + MTX to another b- or ts-DMARDs ± MTX were considered. Both study arms of each RCT were either monotherapy or combination therapy regimens: studies comparing mixed treatment groups or switching to an alternative drug before 24 weeks of treatment were excluded. Therapeutic strategies were then analyzed separately as monotherapy or combination therapy with MTX. The primary efficacy endpoint was the ACR50 response evaluated in both adalimumab and comparison drug arm at 24 weeks of treatment. ACR50 response represents a stringent criterion of disease activity control, radiographic progression and patient satisfaction, assessed in RCTs for RA treatment efficacy [13]. Moreover, ACR50 response has a comparable threshold with low disease activity and/or remission status according to disease activity score routinary assessed in clinical practice [14]. Contextually, the safety endpoint in both arms was the number of withdrawals for any adverse event. In a treat-to-target strategy, EULAR recognizes 24 weeks as the adequate time to reach the target in RA treatment [5] and accordingly we established this time for our comparison.

We quantified the methodological qualities of all studies using Jadad scores [15]. The Jadad scale assesses random assignment, double blinding, and patient withdrawal and dropout rates. Jadad scores range from 0 to 5. Quality was classified as high (score of 3–5) versus low (score of 0–2).

The key words used to evaluate the studies to consider included “rheumatoid arthritis”, “randomized controlled trial”, “head-to-head comparison” and “adalimumab”, or “biologics” or “biological” or “biological drugs” or “small molecules”. Only articles written in English were reviewed. Patients enrolled in all RCTs fulfilled the 2010 ACR/EULAR classification criteria for RA. Finally, according to the meta-analysis protocol, all available RCTs whose treatment arms included targeted therapies, both biologic agents and small molecules, at approved doses for RA were included.

The present systematic review has been registered in International Platform of Registered Systematic Review and Meta-Analysis Protocols (INPLASY) with protocol No. 202220048 (DOI number is 10.37766/inplasy2022.2.0048).

### 2.2. Statistical Analysis and Data Evaluations

Bayesian network meta-analysis (BNM) reflects a prior belief of the possible values of the model parameters of interest, whose likelihood distribution is based on the observed data, and shapes a posterior probability distribution. The reported number of patients in both study arms of every included RCT was used for each analysis. Considering the small number of included studies in the individual arms and to avoid the influence of heterogeneity in demographic characteristics of patients included across the different RCTs with different disease duration and ethnicity, we chose a random-effects model for this network meta-analysis. This method is more appropriate to detect a small real difference and minimize the interference of sample size variability among the different studies analyzed. The results of this MTC are reported as the odds ratio (OR) for a response with every single treatment evaluated head-to-head with the fixed comparator in all RCTs (adalimumab) and the OR for a response emerging from each pairwise combination of the combination treatment (b- or ts-DMARDs ± MTX). Convergence was verified by plots, Monte Carlo error monitoring and with the support of Gelman–Rubin diagnostics, and it was reached at *n*.100,000 iterations. The pair-wise odds ratio (OR) and 95% credible interval (CrI) (or Bayesian CI) and adjusted for multiple-arm trials were reported. Information about the relative effects was converted to a probability and then a ranking of each treatment was presented according to the surface under the cumulative ranking curve (SUCRA), which is expressed as a percentage—the SUCRA would be 100% when a treatment is certain to be the best and 0% when a treatment is certain to be the worst.

The WinBUGS 1.4 software (MRC Biostatistics Unit, Cambridge, UK) was used to combine the direct and indirect evidence of this Bayesian model for this network meta-analysis.

## 3. Results

### 3.1. Studies Selection

A total of 1037 scientific papers were extracted from Embase, Medline, Clinicaltials.gov and Cochrane Library. After a check for duplicates carried out by three authors (F.C.; S.S.; V.V.) independently, 629 papers were screened. The latter abstracts were examined by the same three authors independently, and only RCTs in RA that reported on the use of biologic drugs or small molecules administered at licensed doses with or without MTX in the active arm were retained for further analysis. Among the included abstracts, only 25 research studies presented eligibility data according to the endpoint of interest and the full text of each paper was analyzed by the same three authors independently.

After full paper evaluation, nine studies were excluded: six studies were long-term extension and/or post-hoc analysis of previous published RCTs [16,17,18,19,20,21]; another study was a phase II RCT with a dose-finding endpoint and ACR50 was reported only at 12 weeks [22]. The study from Bernal Rivera L. et al., had only the abstract in English, while the full paper was in Spanish [23]. In the last one adalimumab was compared to baricitinib but a switch to comparator arm was performed before 12 weeks of treatment [24].

Finally, the research articles resulting for eligibility were 16 RCTs in monotherapy or combination therapy, analyzing 8743 RA patients (6484 on combination and 2259 on monotherapy), and were included in the quantitative synthesis of data. A flow chart of the study selection process is shown in Figure 1.

Among studies whose reference treatment was combination therapy of adalimumab originator and methotrexate (MTX), four compared adalimumab to its biosimilars [25,26,27,28], two studies compared adalimumab to tofacitinib [29,30] and one each for adalimumab compared to abatacept [31], certolizumab pegol [32], baricitinib [33], upadacitinib [34] and filgotinib [35]. Among studies whose reference treatment was adalimumab originator monotherapy, three studies compared adalimumab originator to its biosimilar [36,37,38], and one each for tocilizumab [39] and sarilumab [40], respectively. Figure 2 shows the mixed treatment comparison design, in which the reference treatment is adalimumab originator with or without MTX compared to all other targeted therapies (see Figure legend).

### 3.2. Drug Efficacy and Safety Data

The results of the comparison between adalimumab originator and other targeted therapies, combined to MTX, in inducing an ACR50 response are showed in Table 1.

Among combination treatment strategies, upadacitinib ranked first with the most favorable SUCRA to achieve an ACR50 response rate at 24 weeks of treatment, while baricitinib and tofacitinib ranked as second and third, respectively (Table 2). It is relevant that in monotherapy, the adalimumab originator had a more favorable SUCRA for the ACR50 response rate at 24 weeks compared to its biosimilar (Table 2).

Considering the mean posterior probability of reaching ACR50 in combination therapy with MTX, upadacitinib was the best for the probability of being more effective (64%) compared to adalimumab biosimilar, reaching the highest OR among all treatment alternatives, even if statistical significance was not reached (Table 3 top panel). Considering studies in which drugs were used as monotherapy, tocilizumab ranked as first in inducing ACR50 response at 24 weeks of treatment and had a more than two-fold increased probability of being more effective than adalimumab originator and biosimilar. Similarly, sarilumab ranked second and resulted as more effective than both adalimumab originator and biosimilar.

Analyzing the number of patients’ withdrawals due to any adverse event at 24 weeks as safety outcome, among the eight treatments in combination with MTX or the four treatments as monotherapy options none reached statistical significance (Table 3 bottom panel).

## 4. Discussion

Evidence on which therapeutic strategy allows the best outcome to be achieved in RA patients is a hot topic. Unfortunately, head-to-head RCTs often limit themselves to assessing non-inferiority of compared agents, being dominated by industrial interests. Therefore, comparative evaluations are difficult to extend to different drugs, especially when their patent is far from expiring.

The results of the present network metanalysis add some news, giving original insights into the treatment options that may be considered for RA patients.

We conducted a network meta-analysis to compare the probability of achieving ACR50 after 24 weeks of treatment with different targeted therapies and adalimumab as common comparator in head-to-head phase 3 RCTs, both in combination with MTX or as monotherapy, in patients with active RA that had inadequate response to previous csDMARDs. Among all MTX combination treatment options we found data about adalimumab comparison to its biosimilar, abatacept, baricitinib, certolizumab pegol, filgotinib, tofacitinib, upadacitinib. Regarding monotherapy, we found results comparing adalimumab to its biosimilar, sarilumab and tocilizumab.

Previous network meta-analysis assessed efficacy and safety of different targeted therapies in patients with active RA in distinct settings of inadequate response to MTX or anti-TNF agents, comparing biosimilar to its originator [41,42], or comparing different Jak-inhibitors [43], using MTX as common comparator. We used adalimumab as common comparator in a shared setting of MTX-inadequate response in all available head-to-head RCTs.

Compared to traditional meta-analysis, the Bayesian network meta-analysis allows simultaneous comparison of all treatment options. This Bayesian network meta-analysis performed a comprehensive and simultaneous assessment of eight different targeted therapies in combination with MTX and four as monotherapy for active RA showing an inadequate response to previous csDMARDs, and for the first time a TNF-alpha inhibitor was simultaneously compared to other mechanisms of action. Anti-TNF agents, and particularly adalimumab, represent the standard of care for active RA with inadequate response to MTX. Adalimumab was the comparator drug of different head-to-head RCTs included in our meta-analysis, showing almost non-inferiority or even better results in direct comparisons.

Our findings from an indirect comparison suggest that combination of JAK inhibitors and MTX may be more effective in achieving ACR50 after 24 weeks in active RA. Particularly, upadacitinib 15 mg daily ranked as first, followed by baricitinib 4 mg qd, tofacitinib 5 mg twd, and filgotinib 200 qd. Subsequently, certolizumab pegol 200 eow, abatacept 125 mg s.c. weekly, and then adalimumab biosimilar and originator 40 mg eow complete the ranking. These findings may be explained by the different effect of contemporary modulation on multiple pathways of the immune system response or the single antagonism of a specific cytokine. The broader modulation exerted by the inhibition of Janus kinase phosphorylation, and subsequent interfering with different cytokines repertoire, may result in increased efficacy but also may raise concerns about safety.

Instead, the availability of direct comparison as monotherapy is limited to adalimumab originator to its biosimilar, sarilumab or tocilizumab. From these studies we evidenced that IL-6 receptor antagonism, by tocilizumab ranked as first and subsequently by sarilumab, was more effective in achieving ACR50 in active RA patients. Interestingly, adalimumab originator as monotherapy ranked before its biosimilar in the probably to achieve the target.

The unsignificant posterior odds ratios observed in our results could be explained by the statistical design of single considered head-to-head RCT, designed in most cases for non-inferiority between tested agents as a primary endpoint. In previous Bayesian Network meta-analysis, the common comparator that allows indirect comparison between different target therapies was MTX, as standard of care reported as a placebo group. The statistical design of these RCTs was to obtain superiority, thus the results of indirect comparisons reached the statistical significance. Nevertheless, the application of SUCRA allowed a ranking in the probability of reaching ACR50 at 24 weeks of treatment.

Moreover, this Bayesian network analysis strengthens the importance of biosimilars, which showed non-inferiority to their parent drugs, to improve access to biologics and increase sustainability. On the other hand, it demonstrates some differences in terms of efficacy. Moreover, selectivity must be considered when patients affected by active RA non responders to previous DMARDs are evaluated in a clinical scenario where therapy may require progressive step-ups if treatment is not effective within 6 months. Patients not responsive to methotrexate could start the most appropriate combination therapy; those not responsive to any bDMARDs could be swapped or switched to another monotherapy or combination therapy most likely to reach the desired outcome. Our results may be also useful in this scenario.

We decided to assess ACR50 at 24 weeks of treatment to have a target that could be as close as possible to clinical practice in real-life settings and at the same time evaluated in different RCTs. Indeed, in the design of some head-to-head RCTs at week 24 patients could be subjected to crossovers, thus eliminating the possibility of further trusted indirect comparisons.

We observed that none of the targeted therapies included in our Bayesian metanalysis were associated with significant higher risk of withdrawal due to adverse events, as a surrogate of safety. Nevertheless, the common follow-up period of RCTs to investigate efficacy through a direct comparison of drugs unfortunately may not be enough to adequately judge safety with biologic or target synthetic drugs, including cardiovascular events or malignancies, which are important safety issues. Longer comparative studies from multi-center registries represent a more appropriate way of assessing safety concerns.

Some limitations of our study must be considered. The included studies have heterogeneous characteristics and are mostly statistically designed to evaluate non-inferiority between competitor drugs. Indeed, the choice to compare efficacy at 24 weeks and considering only ACR50 was made to minimize the heterogeneity between the 16 studies from different part of the world.

The narrow inclusion criteria for the RA-approved dosage of different included drugs that have the possibility to adapt the dose to optimize efficacy or safety in specific categories of RA patients have led to the exclusion of some studies from the analysis. Although the possible impact of patients’ clinical-demographic differences from RCTs that are functionally unalike has been minimized using the random effect, the extrapolation from different populations to others would be still limited. Finally, some evaluated drugs were only tested in a RCT with a relatively small number of enrolled patients.

In conclusion, although patients with active RA and inadequate response to previous MTX have different therapeutic options of targeted therapies, there are some differences in terms of probability of achieving an ACR50 response after 24 weeks of treatment and these findings may help to define the therapeutic approach in RA patients that respond inadequately or are intolerant to MTX.

The use of a JAK inhibitor in combination with MTX or tocilizumab as monotherapy may represent valid alternatives with higher posterior probability to achieve ACR50 response after 24 weeks. No significant differences in withdrawals due to adverse events within 24 weeks of treatment have been observed. From an economic viewpoint, the only clear advantage that adalimumab has over all other agents mentioned in our study is its cost, due to the availability of biosimilars. Therefore, the justification for starting another target therapy in a RA patient, before having tried adalimumab, can only be justified if a personalized analysis recognizes some peculiar patient circumstances (such as fear of needles, need to travel or difficulty in keeping the drug at a low temperature, etc.), or some condition that can predict anti-TNF agents’ failure (such as use in monotherapy).

Nevertheless, we believe that the availability of a ranking of targeted therapies in RA patients’ treatment may help physicians in clinical practice to personalize treatment strategy.

## Figures and Tables

**Figure 1 jpm-12-00353-f001:**
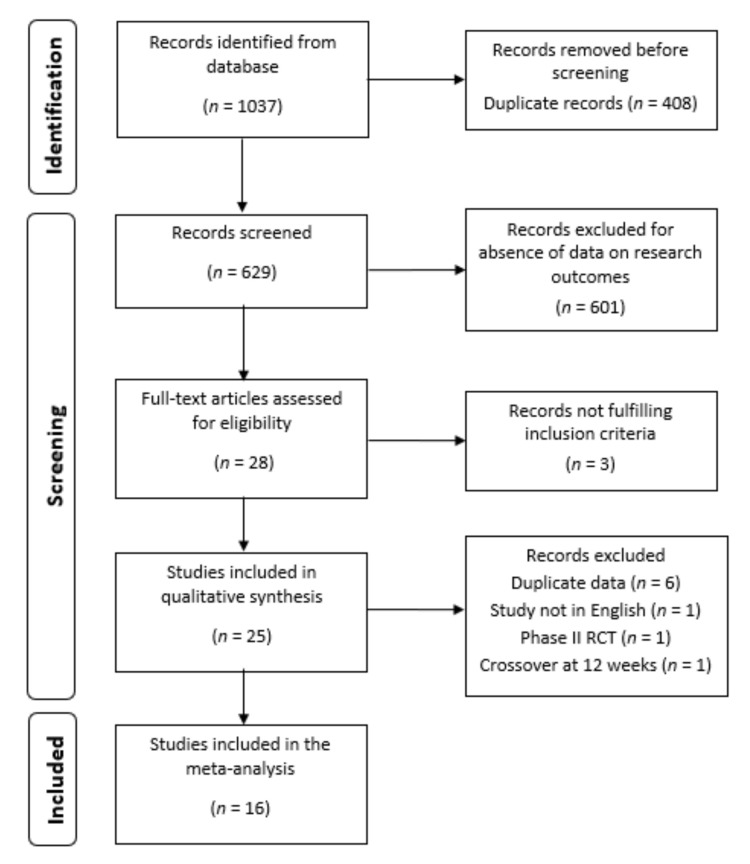
Study flow chart of the article selection process. RCT, randomized controlled trial.

**Figure 2 jpm-12-00353-f002:**
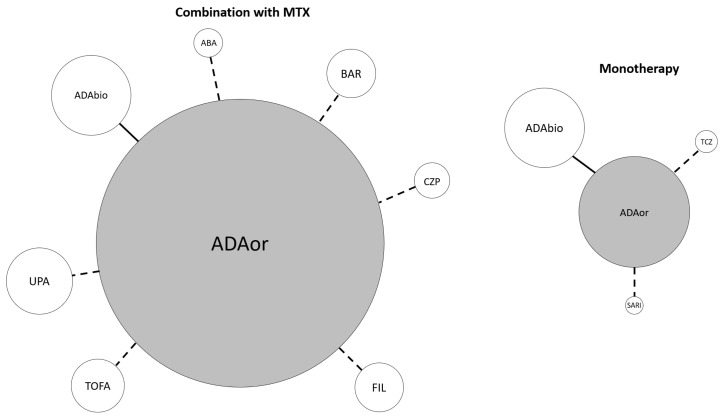
Mixed treatment comparison design. Reference treatment is adalimumab originator ± methotrexate. The continuous connection line means that two or more RCTs compared the connected items. The dotted connection line indicates that a single RCT evaluating the specified comparison was included. The width of each circle is proportional to the cumulative amounts of patients randomized for the specified agent. ABA abatacept, ADA adalimumab, BAR baricitinib, bios biosimilar, CZP certolizumab pegol, FIL filgotinib, MTX methotrexate, or originator, SARI sarilumab, TCZ tocilizumab, TOFA tofacitinib, UPA upadacitinib.

**Table 1 jpm-12-00353-t001:** Characteristics of individual studies included in the network meta-analysis.

Study	Country	Tot. N. Patients	Agent	N. Patients Each Arm	ACR50 % at 24 Weeks	Drop-Out (%)	Jadad Score
Combination therapy with MTX							
Jamshidi, 2017 [25]	Iran	128	ADAbio	64	76.6	4 (6.3)	3
			ADAor	64	75.0	4 (6.3)	
Fleishmann, 2018 [26]	Worldwide	559	ADAbio	286	56.3	44 (15.4)	4
			ADAor	273	52.7	78 (28.6)	
Genovese, 2019 [27]	Worldwide	661	ADAbio	333	48.9	34 (10.2)	4
			ADAor	328	49.4	35 (10.7)	
Sinha, 2020 [28]	India	159	ADAbio	107	80.4	5 (4.7)	4
			ADAor	52	80.8	4 (7.7)	
van Vollenhoven, 2012 [29]	Worldwide	312	TOFA	150	40.7	54 (36)	4
			ADAor	162	32.1	42 (25.9)	
Fleischmann, 2017 [30]	Worldwide	762	TOFA	376	46.0	73 (19.4)	4
			ADAor	386	43.8	74 (19.2)	
Weinblatt, 2013 [31]	North and South America	594	ABA	296	41.2	NA	3
			ADAor	298	39.9	NA	
Smolen, 2016 [32]	Worldwide	714	CTZ	353	64.0	NA	3
			ADAor	361	62.6	NA	
Taylor, 2017 [33]	Worldwide	817	BAR	487	50.5	24 (4.9)	4
			ADAor	330	45.5	7 (2.1)	
Fleischmann, 2019 [34]	Worldwide	978	UPA	651	54.1	20 (3.1)	5
			ADAor	327	41.9	12 (3.7)	
Combe, 2021 [35]	Worldwide	800	FIL	475	57.9	50 (10.5)	4
			ADAor	325	53.8	36 (11.1)	
Monotherapy							
Cohen, 2017 [36]	EU, North and Latin America	496	ADAbio	244	49.2	21 (8.6)	4
			ADAor	252	52.0	11 (4.4)	
Weinblatt, 2018 [37]	EU	476	ADAbio	239	38.1	17 (7.1)	4
			ADAor	237	39.7	19 (8)	
Cohen, 2018 [38]	Asia, EU, USA, Latin America	593	ADAbio	298	36.9	26 (8.7)	3
			ADAor	295	35.9	26 (8.8)	
Gabay, 2013 [39]	North and South America, Australasia, EU	325	TCZ	163	47.2	24 (14.7)	4
			ADAor	162	27.8	28 (17.3)	
Burmester, 2017 [40]	Worldwide	369	SARI	184	45.7	19 (10.3)	4
			ADAor	185	29.7	28 (15.1)	

(ABA: abatacept; ADAbio: adalimumab biosimilar; ADAor: adalimumab originator; BAR: baricitinib; CTZ: certolizumab pegol; EU: Europe; FIL: filgotinib; MTX: methotrexate; SARI: sarilumab; TOFA: tofacitinib; TCZ: tocilizumab; USA: United States of America; UPA: upadacitinib).

**Table 2 jpm-12-00353-t002:** Rank probability of biologics and target synthetic DMARDS in terms of efficacy based on the number of patients who achieved an American College of Rheumatology 50 response rate as combination therapy with methotrexate or as monotherapy (SUCRA—Surface Under the Cumulative RAnking curve).

	SUCRA
Combination therapy with MTX	
Upadacitinib 15 mg qd	0.8871
Baricitinib 4 mg qd	0.5954
Tofacitinib 5 mg twd	0.5845
Filgotinib 200 mg qd	0.5431
Certolizumab pegol 200 mg eow	0.3888
Abatacept 125 mg s.c. weekly	0.3846
Adalimumab biosimilar 40 mg eow	0.3675
Adalimuamb originator 40 mg eow	0.2491
Monotherapy	
Tocilizumab 8 mg/kg e4w	0.8511
Sarilumab 200 mg eow	0.7497
Adalimumab originator 40 mg eow	0.2324
Adalimumab biosimilar 40 mg eow	0.1668

(qd: quotidian; twd: twice a day; eow: every other week; e4w: every 4 weeks).

**Table 3 jpm-12-00353-t003:** Mean posterior probability of Efficacy (top panel) and Safety (bottom panel) in combination therapy with methotrexate or as monotherapy at 24 weeks of treatment for each agent.

Efficacy (ACR50 Response)							
Combination therapy							
ADAor							
1.05 (0.73–1.50)	ADAbio						
1.06 (0.56–1.97)	1.00 (0.49–2.06)	ABA					
1.23 (0.67–2.25)	1.17 (0.58–2.36)	1.16 (0.49–2.78)	BAR				
1.06 (0.58–1.95)	1.01 (0.50–2.06)	1.01 (0.42–2.42)	0.86 (0.37–2.04)	CZP			
1.21 (0.79–1.95)	1.14 (0.66–2.10)	1.14 (0.54–2.54	0.98 (0.47–2.15)	1.14 (0.55–2.50)	TOFA		
1.64 (0.89–2.99)	1.55 (0.77–3.15)	1.55 (0.65–3.70)	1.33 (0.57–3.12)	1.54 (0.65–3.63)	1.36 (0.62–2.80)	UPA	
1.18 (0.65–2.18)	1.12 (0.56–2.28)	1.12 (0.47–2.68)	0.96 (0.41–2.27)	1.12 (0.47–2.64)	0.98 (0.45- 2.04)	0.72 (0.31–1.70)	FIL
Monotherapy							
ADAor							
0.96 (0.52–1.75)	ADAbio						
2.35 (0.80–6.80)	2.45 (0.71–8.43)	TCZ					
2.00 (0.68–5.79)	2.08 (0.61–7.08)	0.85 (0.18–3.89)	SARI				
SAFETY (withdrawals for any adverse event)							
Combination therapy							
ADAor							
0.88 (0.62–1.20)	ADAbio						
0.85 (0.45–1.43)	1.02 (0.47–1.80)	ABA					
1.22 (0.63–2.05)	1.44 (0.66–2.61)	1.70 (0.62–3.18)	BAR				
1.12 (0.60–1.86)	1.31 (0.64–2.40)	1.49 (0.59–3.03)	1.03 (0.42–2.17)	CZP			
0.90 (0.57–1.38)	1.06 (0.60–1.79)	1.22 (0.54–2.28)	0.83 (0.37–1.62)	0.90 (0.40–1.78)	TOFA		
1.23 (0.66–2.05)	1.44 (0.70–2.64)	1.65 (0.63–3.42)	1.13 (0.44–2.38)	1.23 (0.50–2.54)	1.19 (0.38–1.59)	UPA	
1.20 (0.64–2.01)	1.42 (0.68–2.62)	1.63 (0.63–3.20)	1.11 (0.44–2.31)	1.19 (0.49–2.38)	1.40 (0.63–2.63)	1.11 (0.43–2.29)	FIL
Monotherapy							
ADAor							
0.91 (0.49–1.53)	ADAbio						
1.10 (0.32–2.64)	1.42 (0.32–3.59)	TCZ					
1.20 (0.39–2.71)	1.51 (0.37–3.71)	1.70 (0.26–4.74)	SARI				

Odds ratio < 1 means that treatment intercepting on the right is more effective or safer. (ABA: abatacept; ADAbio: adalimumab biosimilar; ADAor: adalimumab originator; BAR: baricitinib; CTZ: certolizumab pegol; FIL: filgotinib; SARI: sarilumab; TOFA: tofacitinib; TCZ: tocilizumab; UPA: upadacitinib).

## Data Availability

The data presented in this study are available in respective studies published and reported within the article references. No new data were created or analyzed in this study.

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
