# Peer review of "Comparison of Adalimumab to Other Targeted Therapies in Rheumatoid Arthritis: Results from Systematic Literature Review and Meta-Analysis"

_jpm, 2022, doi:10.3390/jpm12030353_

Round 1

Reviewer 1 Report

The authors have not addressed my previous comments. So, I still reject this manuscript.

Reviewer 2 Report

I would like to thank authors for their review

Reviewer 3 Report

All my queries were answered.

This manuscript is a resubmission of an earlier submission. The following is a list of the peer review reports and author responses from that submission.

Round 1

Reviewer 1 Report

This manuscript describes a meta-analysis of different targeted RA therapies, compared to Adalimumab, either alone or in combination with methotrexate (MTX). The study finds that in combination with MTX, Adalimumab is likely to be inferior to JAK inhibitors, CTLA4 activation (Abatacept), and to another TNF inhibitor (Certolizumab); as monotherapy, Adalimumab is likely to be inferior to IL-6 inhibitors.  

As far as this reviewer is concerned, only the possible inferiority of Adalimumab to Certolizumab and Abatacept can be considered a surprise, and these findings do not appear to be particularly robust (the difference in SUCRA is low). All other findings are - at least in my experience - well-established clinical knowledge. Hence, the clinical interest of the manuscript is quite low.

Also, the authors fail to address properly the one clear advantage Adalimumab has over all other agents mentioned in the study: its cost. Due to the availability of biosimilars (discussed at some length in the manuscript), Adalimumab is considerably less costly than the alternatives. Therefore, the justification for starting another tDMARD in a patient, before having tried Adalimumab (or Etanercept), can only be justified from an economic viewpoint if a personalized analysis can predict treatment failure of Anti-TNF agents. In this sense, this manuscript provides no additional information.

Therefore: although the paper is well written, the analysis properly performed (as far as I can judge), and the subject has some clinical relevance, I do not recommend publication. The manuscript lacks one central aspect of scientific work: new information.

Reviewer 2 Report

This study is a meta-analysis that evaluated the effectiveness of some bDMARD or tsDMARD with comparing adalimumab.

  1. Why did theauthors compare the effectiveness of drugs with adalimumab?
  2. The included studies have heterogeneous characteristics. This study is a meta-analysis not a randomised controlled study. Therefore, the results have no certainty. The statement “Our findings suggest that combination of JAK inhibitors and MTX is the most effective treatment for achieving ACR50 after 24 weeks in active RA.” should be revised. Additionally, ranking the effectiveness of the drugs based upon the results may not be valid. Because most of the included studies were designate to evaluate non-inferiority.
  3. The authors included 16 studies from different part of the world. Additionally, most of the efficacy of drugs has not been evaluated in more than one study.
  4. Did only criteria related to patients for including study is presence of active disease.
  5. The authors did not mention about side effects of the drugs in the study.

Reviewer 3 Report

Peer review report on ‘Comparison of adalimumab to other targeted therapies in rheumatoid arthritis : results from systematic literature review and meta-analysis

  1. Original submission

Recommendation

Reject

  1. Comments to authors

Ms. Ref. No. : jpm-1555239

Title : Comparison of adalimumab to other targeted therapies in rheumatoid arthritis : results from systematic literature review and meta-analysis

Overview and general recommendation

The authors wished to assess the efficacy and safety of adalimumab in comparison to other targeted therapies for the treatment of rheumatoid arthritis (RA) via a Bayesian network meta-analysis. However, they did not reach their objective and the manuscript is full of several biases in the methodology and data interpretation. Furthermore, there is no clear clinical relevance for comparing adalimumab to other targeted therapies. I thererefore recommend rejection of this manuscript.

Major comments

  • Please, revise to implement all the changes from the full-text. As per the journal’s guidelines for mansucript writing, make an up to 200 words abstract, and in a single paragraph following the style of structured abstracts, but without subtitles.
  • Please, consider reducing the length of the introduction to three paragraphs that cover at most one page and revising its content : first paragraph including information on available rheumatoid arthritis (RA) treatments and treatment recommendations, second paragraph about current unmet needs in the treatment of rheumatoid arthritis (notably when csDMARDS have failed) including how and why adalimumab (originator) is different from other targeted therapies, third paragraph about the study aim clearly stated. It is unclear why you compare adalimumab to othet targeted therapies. What is the clinical relevance of such a comparison ?
  • The current methods section is disordered. For more clarity, I suggest specifying the content of the Methods section within these subtitles : data source, study selection (inclusion and exclusion criteria), data extraction, appraisal of individual study quality and statistical analysis. Page 2, lines 92-93 : the search strategy is incomplete, with words such as « targeted therapies » or « biologicals » or « biologics » or « biological drugs » or « small molecules » missing. Why did you exclude RA patients classified with 1987 ACR criteria meanwhile your search period was from database inception up to june 2021 (this increases the selection bias)? Why did you excluded studies published in languages other than English whereas there are online translator tools (this further increases the selection bias)? Based on your results from table 2, SUCRA is likely useful to rank treatment efficacy, but not to compare the efficacy of different treatments to that of a reference one as you are supposed to do in this study. The relevance of SUCRA here is therefore questionable. Page 3, lines 119-120 : if SUCRA can compare treatment efficacy with reference to a standard treatment, which SUCRA values show that a treatment is better than the reference treatment, even if it is not the best one ? Why did you not choose to assess efficacy and safety in a longer term (at 52 weeks for example) to better assess long-term efficacy and safety data? Why did you not assess secondary outcomes such as ACR 70 which shows better efficacy? Did you ensure transitivity and consistency that are important determinants of the validity of your results?
  • Page 3, line 125 : I suggest changing the subtitle « Studies included in the analysis » to « 3.1. Study selection » and to summarize informations from this subsection in a unique paragraph. I also suggest two other subtitles : « 3.2. Drug efficacy data » and « 3.3. Drug safety » instead of mixing both groups of results under the same subtitle « Network meta-analysis of efficacy and safety at 24 weeks of treatment». Please, revise figure 1 to make it more comprehensive. Take a PRISMA flow chart here http://prisma-statement.org/prismastatement/flowdiagram.aspx and fill it. Please, provide general characteristics of included studies in a Table that you can put in supplementary materials. Results from table 3 show that no assessed treatment was probably more effective than adalimumab originator combination therapy with methotrexate or adalimumab monotherapy, and the interpretation bias in page 6, lines 181-184 should be corrected. Page 7, lines 197-199 : this sentence should be revised to say that based on results from table 4, there was probably no difference between adalimumab and other targeted therapies in terms of side effects. Which side effects did you find ? Can you summarisze them in a table or figure (with respective frequencies)?
  • The current discussion is completely out of the topic, and inconsistent with your study aim. Especially, instead of discussing results of efficacy and safety of drugs in comparison to adalimumab, you discuss ranking of all treatments efficacy.
  • The conlcusion should be revised to fit with the study aim.

1.7.References are not well formatted according to the journal’s guidelines.

.

Minor comments

  1. Page 3, lines 127-129 : specify the authors who removed duplicates and selected the articles.
  2. Page 4, lines 147-153 « Among the studies….and sarilumab (SAR) [38], respectively. » I suggest rephrasing like this : « Among studies whose reference treatment was combination therapy of adalimumab originator and methotrexate, four studies compared adalimumab to its biosimilar, two studies compared adlimumab to tofacitinib, and one each for adalimumab compared to abatacept, certolizumab pegol, baricitinib, upadacitinib and filgotinib. Among studies whose reference treatment was adalimumab originator monotherapy, three studies compared adalimumab to its biosimilar, and one each for tocilizumamb and sarilumab. »
  3. Please, consider removing the assumption « Pooled results were considered statistically significant if the 95% CI did not contain the value 1 » from page 5, lines 115-116 which is unnecessary.
  4. In table 1, please change the word « Agent » to « Drug ». Tofacitinib is usually abbreviated « TOFA » instead of TOF. Sarilumab is usually abbrviated « SARI ». For the study of Smolen 2016, is it TCZ or CTZ ?
  5. Please, avoid abbreviating names of drugs throughout the text. Rather keep abbreviations for figures and tables.
  6. Page 6, lines 172-174 : Please, clarify that sentence.
  7. Page 7, lines 194-195 : this comment about SUCRA should be mentioned above where results from table 2 are presented.
  8. Typos need to be corrected, e.g. Page 8, line 241 : probability instead of propbably.

Anonymous

Available online 12 January 2022

Reviewer 4 Report

This paper has a highly relevant research question, well-design and appropriate handling of the data. I have some questions and suggestions.

  • As already stated as a limitation, please broaden the discussion regarding the heterogeneities in included studies. eg. ACR50 response of 80.4% vs 46% in different studies. Please discuss the possible reasons and effects of this issue on the conclusions of this current study.
  • There are some typo errors need correction. eg. ''Pubblications'' in Figure 1.
  • In Figure 2, thick and thin lines can not be seen clearly. I suggest to find another way to visualize this lines.